# Research Progress in Semiconductor Materials with Application in the Photocatalytic Reduction of CO$_2$

**Yan Cui, Pengxiang Ge, Mindong Chen \* and Leilei Xu**

Jiangsu Key Laboratory of Atmospheric Environment Monitoring and Pollution Control, Collaborative Innovation Center of Atmospheric Environment and Equipment Technology, School of Environmental Science and Engineering, Nanjing University of Information Science & Technology, Nanjing 210044, China; cuiyan@nuist.edu.cn (Y.C.); gepx@nuist.edu.cn (P.G.); leileixu88@gmail.com (L.X.)
\* Correspondence: chenmd@nuist.edu.cn; Tel.: +86-25-5873-1089

**Abstract:** The large-scale burning of non-renewable fossil fuels leads to the gradual increase of the CO$_2$ concentration in the atmosphere, which is associated with negative impacts on the environment. The consequent need to reduce the emission of CO$_2$ resulting from fossil fuel combustion has led to a serious energy crisis. Research reports indicate that the photocatalytic reduction of CO$_2$ is one of the most effective methods to control CO$_2$ pollution. Therefore, the development of novel high-efficiency semiconductor materials has become an important research field. Semiconductor materials need to have a structure with abundant catalytic sites, among other conditions, which is of great significance for the practical application of highly active catalysts for CO$_2$ reduction. This review systematically describes various types of semiconductor materials, as well as adjustments to the physical, chemical and electronic characteristics of semiconductor catalysts to improve the performance of photocatalytic reduction of CO$_2$. The principle of photocatalytic CO$_2$ reduction is also provided in this review. The reaction types and conditions of photocatalytic CO$_2$ reduction are further discussed. We believe that this review will provide a good basis and reference point for future design and development in this field.

**Keywords:** semiconductor materials; photocatalytic; carbon dioxide; catalysts

## 1. Introduction

Over the past hundred years, the establishment and development of industrial civilization has promoted a profound transformation of human social productivity and created huge social wealth for mankind. In the process of industrial development, the annual increase in the consumption of fossil fuels, an important energy source for industrial development, has led to the degradation of the natural carbon cycle as well as a surge in atmospheric CO$_2$ concentration [1–4]. Especially in recent decades, due to the rapid population increase and the rapid development of industry, the CO$_2$ produced by human respiration and fossil fuel combustion has far exceeded historical levels [5]. In addition, human development has led to ecological destruction, weakening the ability of Nature to absorb CO$_2$ and resulting in increasing levels of CO$_2$ in the atmosphere.

In recent years, studies on CO$_2$ fixation and transformation have gradually developed. Adsorption, storage, reduction and conversion methods can reduce atmospheric CO$_2$ concentration [6]. However, these methods achieve little, and they do not solve the fundamental problem. CO$_2$ is a potential carbon resource, and its utilization is a promising way to reduce CO$_2$ to fuel. Therefore, strategies for CO$_2$ conversion have been developed. These reduction pathways include electrochemical reduction, photochemical reduction, and biotransformation [7,8]. Photochemical reduction is the most promising method for CO$_2$ conversion. It converts Nature's nearly unlimited solar energy into usable chemical energy, achieving the conversion and utilization of CO$_2$ without consuming fossil energy. At the same time, solar energy is a clean and renewable resource.

In view of this, the conversion of $CO_2$ to hydrocarbons is considered a forward-looking solution because it can solve both the energy problem and the environmental crisis [9–13]. In this regard, artificial photocatalytic reduction is one of the most promising solutions, and the use of this method to photocatalytically reduce $CO_2$ into usable fuels has attracted special attention worldwide [14]. Various semiconductors, including $TiO_2$ [15–17], MOFs [18–20], $BiVO_4$ [21–23] and metal halide perovskites (MHPs) [24–26], have been used as photoreduction catalysts for $CO_2$. Although these semiconductor photocatalysts have a wide band gap and can provide enough negative electrons for the photocatalytic reduction of $CO_2$, most of them only respond to ultraviolet light (UV), which severely limits their use of light. In addition, these semiconductor photocatalysts usually have low photocarrier separation and migration efficiency [27,28]. These problems greatly reduce the efficiency of photocatalytic $CO_2$ reduction. Take the most commonly used $TiO_2$ photocatalyst as an example. $TiO_2$ photocatalysts have low electron mobility and poor adsorption activation capacity for $CO_2$ reactants [29]. Therefore, much research has been devoted to various modifications to improve the efficiency of photocatalytic $CO_2$ reduction. The transition metals cobalt and nickel are often used in the study of photocatalytic $CO_2$ reduction due to their rich crustal content, high stability, low cost and high catalytic activity. Cobalt and nickel metal ions can also effectively avoid the generation of high-energy intermediates and promote the multi-electron reduction of $CO_2$, so they are favorable components in photocatalytic $CO_2$ reduction processes [30–32]. It has been reported that defect engineering can change the band structure and optical properties of semiconductor photocatalysts, thus improving the light absorption capacity of semiconductors and improving the carrier separation efficiency [33–35]. As one of the most efficient solutions, surface modification can optimize the surface atomic states, surface structures and charge properties of semiconductor photocatalysts, and is therefore a promising photocatalytic reaction technology [36–38]. Vu et al. [36] reviewed different oxide catalysts for $CO_2$ reduction together with various structural engineering strategies (e.g., nanostructure engineering and heterostructure engineering). Each strategy was found to have its own advantages and disadvantages, so further adjustment is needed to further improve the performance of photocatalysts. For example, photocatalysts are modified with precious metals. These precious metals can play the role of modifier and provide more catalytic active sites for the photocatalytic reduction of $CO_2$. Although these precious-metal modifiers can significantly improve the photoreduction performance of $CO_2$, their rarity and high cost greatly affect their practical application [39–42]. Therefore, it is necessary to develop non-noble metal or no-metal modified photocatalysts. Given the challenges and limitations of current solutions for photocatalytic $CO_2$ reduction, it is of great significance to design novel photocatalysts to improve the photoreduction of $CO_2$.

However, existing reviews on the photocatalytic reduction of $CO_2$ by semiconductor materials are not comprehensive, and the recent research progress has not been sufficiently summarized. Therefore, this review summarizes the progress of research on semiconductor catalysts which have recently demonstrated excellent performance. In this paper, we review the effects of different types of semiconductor catalysts on the photocatalytic reduction of $CO_2$. In addition, we summarize the mechanisms of photocatalytic $CO_2$ reduction on these semiconductor catalyst. Finally, we discuss future development trends of semiconductor catalysts.

## 2. Semiconductor Materials for the Photocatalytic Reduction of $CO_2$

The photocatalytic reduction of $CO_2$ is a surface/interface reaction. As discussed above, it is important to find and use raw materials which are environmentally friendly and effective as catalysts. In 1979, Inoue et al. [43] used semiconductor materials such as CdS, $TiO_2$ and $WO_3$ to reduce $CO_2$ to CO, $CH_4$ and other chemical. Figure 1 shows the band structures of different semiconductor photocatalysts. Currently known semiconductor photocatalyst materials for the photocatalytic reduction of $CO_2$ include metal oxides ($Co_3O_4$, $BiVO_4$, $TiO_2$), metal chalcogenides (CdS, $MoS_2$, $SnS_2$), nitrides (g-$C_3N_4$), bismuth halide

oxides (BiOCl, BiOBr) and bimetallic hydroxides (NiAl-LDH), among others. Although the photocatalysts mentioned above have been proven able to reduce $CO_2$, they still have some disadvantages. For example, bismuth halide compounds have strong oxidation ability, but the bottom position of their conduction band is comparatively positive, and their reduction performance is not very good [44,45]. Some photocatalysts have unstable chemical properties and are prone to photocorrosion during photocatalytic reaction, the products of which are toxic to the environment.

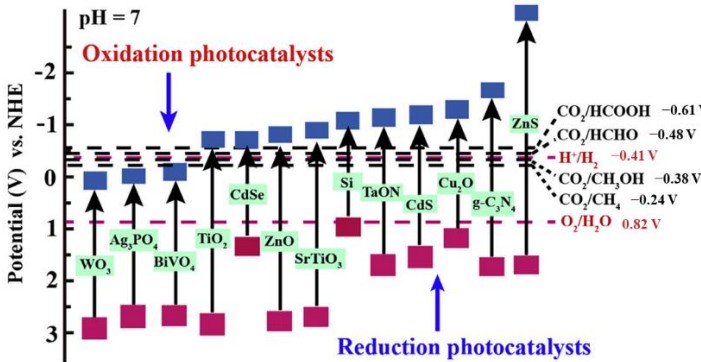

**Figure 1.** Band structures of some representative photocatalysts. Reprinted with permission from Ref. [46]. 2020 Xu, Q.; Zhang, L.; Cheng, B.; Fan, J.; Yu, J.

### 2.1. TiO₂ Photocatalyst

As an n-type semiconductor material, $TiO_2$ is regarded as an ideal semiconductor material for environmental pollution control due to its good chemical stability, non-toxicity and environmental friendliness. It has great potential application value in the field of environment and energy [47]. In 1972, Fujishima and Honda reported [14] that $TiO_2$ could decompose water molecules into hydrogen and oxygen under ultraviolet light, bringing extensive attention to $TiO_2$ as a photocatalyst material. $TiO_2$ has been widely used in the photocatalytic decomposition of decomposition of hydrogen in aquatic products [48–50], the degradation of pollutants [51–53], the reduction of $CO_2$ [54–57] etc., and a series of research achievements have been made. $TiO_2$ photocatalyst materials are also very common in daily life; they are widely used in solar cells, coatings, cosmetics, antibacterial materials and air purifiers, as examples.

$TiO_2$ can be found in the form of three crystal polymorphs: anatase, rutile and brookite [58]. The three crystals have a twisted octahedral structure of six oxygen atoms around a titanium atom. Anatase and rutile forms have tetragonal crystal structures. The different electronic structures of the three crystal types of $TiO_2$ lead to great differences in their photocatalytic performance. Tang et al. [59] studied the effect of different crystal types on the photocatalytic performance of $TiO_2$. It was found that the degradation rate of pollutants was nearly 100% when anatase or mixes with anatase and rutile forms was used as photocatalyst. The degradation rate was less than 15% when pure rutile $TiO_2$ was used as photocatalyst. Jin et al. [60] prepared PbO-decorated $TiO_2$ composites by a one-pot method with highly photoactive $CO_2$ conversion, as shown in Figure 2. The heterojunction formed by the catalyst could effectively inhibit the recombination of photogenerated charge, while the PbO could improve the adsorption of $CO_2$ on the catalyst. Therefore, the photocatalytic activity of the heterojunction complex for $CO_2$ reduction was significantly improved.

Morphology is also an important factor affecting the photocatalytic performance of $TiO_2$. The specific surface areas, active sites, charge transfer rates and exposed crystal surfaces of $TiO_2$ catalysts with different morphologies are significantly different, leading to great differences in their performance. Cao et al. [61] prepared a $TiO_2$ photocatalyst with nanorods and nanorod-hierarchical nanostructures. The catalyst had better photocatalytic performance for $CO_2$ reduction than commercial $TiO_2$ (P25). The high catalytic activity was mainly attributed to the improved charge transfer performance, specific surface area and

light absorption performance of the catalyst lent by the nanorod-hierarchical nanostructures. Tan et al. [62] synthesized an Ag/Pd bimetal supported on a N-doped $TiO_2$ nanosheet for $CO_2$ reduction. Due to the modification of the Ag/Pd bimetal and the N doping, the absorption of visible light in the $TiO_2$ nanosheets was improved. This system also provided abundant surface defects and oxygen vacancies for the $TiO_2$ nanosheets, causing the catalyst to exhibit high performance for the photocatalytic reduction of $CO_2$ to $CH_4$. Kar et al. [63] prepared a $TiO_2$ nanotube photocatalyst which showed a highly efficient photocatalytic reduction of $CO_2$ to $CH_4$. The high photocatalytic activity was mainly attributed to the enhancement of visible light absorption by the nanotube structure. In addition to the crystal size and morphology of $TiO_2$ catalyst, the intensity of the external light source also has an effect on the photocatalytic activity of $TiO_2$—generally, as the light intensity increases, more photogenerated electrons are generated by the catalyst's excitation, and the photocatalytic reaction is promoted as a result. Yang et al. [64] found that the photocatalytic degradation rate of $TiO_2$ to paracetamol attenuated with the decrease of light intensity.

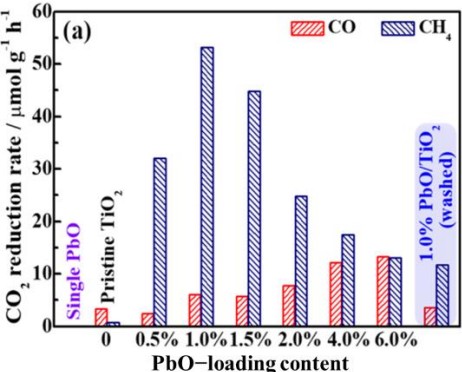

**Figure 2.** Photocatalytic CO/$CH_4$ production activities of the single PbO, pristine $TiO_2$ and PbO/$TiO_2$ composites. Reprinted with permission from Ref. [60]. 2020 Jin, J.; Chen, S.; Wang, J.; Chen, C.; Peng, T.

### 2.2. Co/Ni-Based Catalysts

The transition metals cobalt and nickel, which have various redox states, have been widely used in the study of photocatalytic $CO_2$ reduction due to their rich crustal content, low cost, high catalytic activity for $CO_2$ and strong adsorption capacity [65–67]. In 2015, Wang et al. [68] prepared Co-ZIF-9 by a solvothermal method for the study of photocatalytic $CO_2$ reduction. The results showed that the three-dimensional MOFs structure of Co-ZIF-9 was beneficial for $CO_2$ enrichment. The $Co^{2+}$ ions and imidazole groups played a synergistic role in the photocatalytic reduction of $CO_2$. The $Co^{2+}$ ion was beneficial for electron transport, and the imidazole group was beneficial for the activation of $CO_2$ molecules. Wang et al. [69,70] also studied the photocatalytic $CO_2$ reduction performance of cobalt-based spinel oxides. $MnCo_2O_4$ microspheres and $ZnCo_2O_4$ nanorods were prepared by solvothermal calcination and hydrothermal methods, respectively. The results showed that both catalysts exhibited excellent catalytic stability, which confirmed the possibility of Co-based spinel oxides for photocatalytic $CO_2$ reduction. Zhang et al. [71] studied the performance of six Co-MOFs with different coordination environments applied to photocatalytic $CO_2$ reduction, as shown in Figure 3. In pure $CO_2$ atmosphere, the MAF-X27*l*-OH material had high CO selectivity (98.2%). When the relative pressure of $CO_2$ dropped to 0.1 atm, the conversion rate of MAF-X27*l*-OH remained at about 80%, while the CO conversion rate of Co-MOF materials without this ligand decreased significantly. These results suggest that MAF-X27*l*-OH has excellent photocatalytic $CO_2$ reduction performance. These conclusions provide a theoretical basis for future studies on the photocatalytic $CO_2$ reduction of Co-based nanocatalysts.

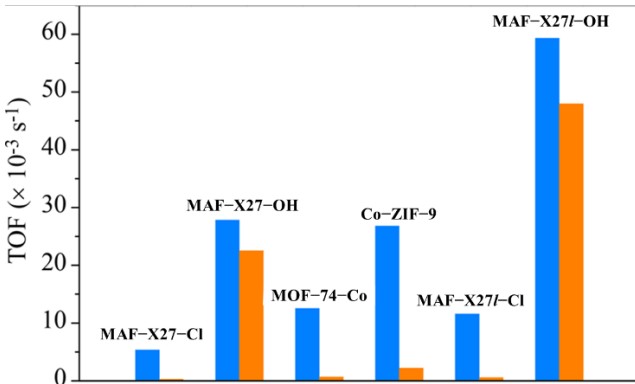

**Figure 3.** TOF values for the photocatalytic reduction of $CO_2$ to CO at 1.0 atm (blue) and 0.1 atm (orange). Reprinted with permission from Ref. [71]. 2018 Wang, Y.; Huang, N.-Y.; Shen, J.-Q.; Liao, P.-Q.; Chen, X.-M.; Zhang, J.-P.

In addition to Co-based nanomaterials, many Ni-based nanomaterials have also been applied in the study of photocatalytic $CO_2$ reduction. Niu et al. [72] synthesized a spongy nickel-organic heterogeneous photocatalyst (Ni(TPA/TEG), which could effectively adsorb $CO_2$. This novel Ni-based photocatalyst significantly inhibited the production of $H_2$, $CH_4$ and $CH_3OH$ during the photocatalytic reduction of $CO_2$. Thus, efficient $CO_2$ conversion was achieved, and the selectivity of CO was close to 100%. Yu et al. [73] obtained a covalent organic framework bearing single Ni sites (Ni-TpBpy) for the selective reduction of $CO_2$ to CO. Figure 4 shows the photocatalytic $CO_2$ reduction performance of Ni-TpBpy under visible light. The results showed that Ni-TpBpy effectively promoted the formation of CO in the reaction medium, and the selectivity of CO was 96% after 5 h of reaction. More importantly, the catalyst maintained 76% CO selectivity in a low $CO_2$ atmosphere. Ni-TpBpy showed good photocatalytic performance for the selective reduction of $CO_2$, which was mainly attributed to the synergistic effect of single Ni catalytic site and TpBpy supporter.

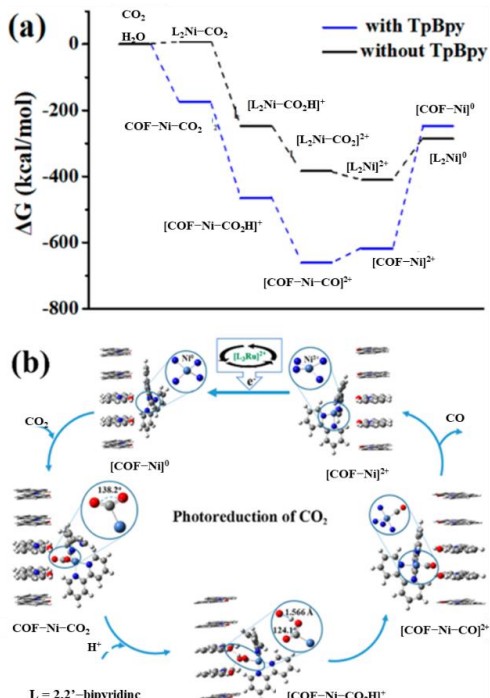

**Figure 4.** (**a**) Comparison of the free energy of $CO_2$ reduction to CO with and without TpBpy ligand; (**b**) proposed mechanism for the photocatalytic $CO_2$ reduction by Ni-TpBpy. Reprinted with permission from Ref. [73]. 2019 Zhong, W.; Sa, R.; Li, L.; He, Y.; Li, L.; Bi, J.; Zhuang, Z.; Yu, Y.; Zou, Z.

### 2.3. Metal Halide Perovskites (MHPs)

In recent years, metal halide perovskite (MHP) materials have become attractive in the field of optoelectronics and energy conversion [74]. Compared with traditional semiconductor nanocrystals, these materials have high extinction coefficient, narrow band emission, long carrier diffusion length and high defect tolerance. In addition, the diversity of perovskite structures also allows the band gap to be adjusted in order to enhance light capture [75]. The crystal structure of MHPs is similar to that of oxide perovskite. The chemical formula is $ABX_3$, where A is a monovalent cation, B is a divalent metal cation (the most common are $Pb^{2+}$ and $Sn^{2+}$) and X is a halogen ion. MHP nanocrystals exhibit a halogen-rich surface structure. Perovskites can be classified as inorganic or organic-inorganic hybrid halogenated perovskites according to the type of cations in their chemical structure. The emergence of MHPs with unique photoelectric characteristics brings new opportunities for efficient photocatalytic $CO_2$ reduction.

Since the reduction potential of MHPs for $CO_2$ reduction usually changes with the nanocrystal size, the catalytic activity of MHP nanocrystals is also affected by their size. Sun et al. [76] first synthesized $CsPbBr_3$ quantum dots of different sizes to study the effect of quantum dot size on $CO_2$ reduction. It was found that $CsPbBr_3$ with a diameter of 8.5 nm had the longest carrier lifetime, more negative band bottom potential and the highest catalytic activity. As shown in Figure 5, Xu et al. [77] designed $CsPbBr_3$/GO composites by combining graphene oxide (GO) with $CsPbBr_3$ and used them for photocatalytic $CO_2$ reduction. The $CsPbBr_3$/GO composites exhibited higher $CO_2$ reduction activity than pure $CsPbBr_3$ nanocrystals. It was found that the improved $CO_2$ reduction performance was mainly attributed to the existence of efficient charge transfer in $CsPbBr_3$/GO composites, and charge injection graphene oxide effectively promoted the charge injection from MHPs to GO. Other studies have shown that the catalytic activity of MHPs can be improved by coupling various modifiers with MHPs nanocrystals. Man et al. [78] used $NH_x$-rich porous g-$C_3N_4$ nanosheets (PCNs) to stabilize $CsPbBr_3$ nanocrystals. The formation of a N–Br bond leads to close contact with g-$C_3N_4$ nanocrystals and $CsPbBr_3$ nanocrystals. In order to suppress the serious charge recombination in MHPs, Jiang et al. [79] designed a novel Z-scheme alpha-$Fe_2O_3$/amine-RGO/$CsPbBr_3$ catalyst for high-efficiency $CO_2$ reduction. The construction of the Z-scheme heterojunction promoted charge separation and retained strong reducing electrons in $CsPbBr_3$ as well as strong oxidation holes in $\alpha$-$Fe_2O_3$. Finally, it promoted the activity of artificial photosynthesis.

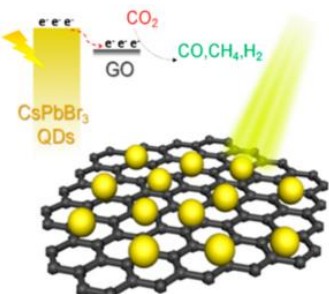

**Figure 5.** Schematic of photocatalytic $CO_2$ reduction over $CsPbBr_3$QDs/GO. Reprinted with permission from Ref. [77]. 2017 Xu, Y.-F.; Yang, M.-Z.; Chen, B.-X.; Wang, X.-D.; Chen, H.-Y.; Kuang, D.-B.; Su, C.-Y.

### 2.4. Metal-Organic Frameworks (MOFs)

Metal-organic framework materials (MOFs) are organic-inorganic hybrid materials with intramolecular pores formed by self-assembly between inorganic metal ions or clusters and organic ligands through coordination bonds. MOFs' large specific surface area, high porosity and tunable structure make them energy storage materials with strong development potential. The structure of MOFs can be controlled by changing the central metal atoms and the interaction between different organic ligands. Transition metals such as Fe,

Co and Ni are often selected as central metal sources. As the raw materials of MOFs, such transition metals are abundant, widely distributed and easily available on Earth, which to some extent makes the raw materials cost of MOFs low. Therefore, MOFs are widely used in catalytic energy conversion and other applications [80]. In photocatalytic processes, MOFs can be used as photocatalysts or modifiers to promote the photocatalytic reaction.

MOFs are a relatively new type of frame material, and their uniformly dispersed metal nodes are conducive to the adsorption and activation of gas molecules. The combination of semiconductors and MOFs to form inorganic-organic nanocomposites can not only satisfy the absorption of photons by semiconductor materials to generate carriers, but also enable the materials to adsorb and activate highly stable $CO_2$ molecules. Xiong et al. [81] developed a method for synthesizing $Cu_3(BTC)_2$@$TiO_2$ core-shell structures. The loose shell structure of $TiO_2$ facilitates the passage of $CO_2$ molecules through the shell, while the strong $CO_2$ absorption of $Cu_3(BTC)_2$ in the core favors the efficient $CO_2$ reduction of nanocomposites. Wu et al. [82] designed and constructed inorganic perovskite quantum dots and organic MOF complexes. A series of MAPbI(3)@PCN-221(Fe-x) composite photocatalytic materials were prepared by encapsulating quantum dots in MOF channels by a deposition method. In the aqueous phase, the MAPbI(3)@PCN-221(Fe-0.2) composite structure showed a high photocatalytic activity for $CO_2$—much higher than that of PCN-221(Fe-0.2). At the same time, hole oxidation oxidized $H_2O$ to $O_2$ to achieve the total decomposition of $CO_2$. Alvaro et al. [83] reported that MOF-5 produced by the coordination of $Zn^{2+}$ and terephthalic acid has semiconductor-like properties. There is an electron-hole separation and transfer process in which electrons migrate from ligand to adjacent metal nodes for the photocatalytic degradation of phenol. Subsequently, a great deal of research was conducted on the construction and development of MOFs with semiconductor properties. On the one hand, MOFs act as light absorbers. On the other hand, due to their unique structural characteristics, they can be used as highly efficient catalysts integrating light absorption and catalytic active sites [84]. Fang et al. [85] designed and synthesized a pyrazolyl porphyrinic Ni-MOF (PCN-601) with active sites of light capture exposure and high specific surface area, as shown in Figure 6. The experimental results showed that PCN-601 had high-efficiency $CO_2$ photoreduction performance under the condition of $CO_2$-saturated water vapor. Under the condition of simulated solar irradiation (AM 1.5 G), the $CH_4$ yield of PCN-601 reached 92 $\mu mol \cdot h^{-1} \cdot g^{-1}$ and the apparent quantum yield (AQY) of $CH_4$ was 2.18%. Therefore, the accumulated research suggests that MOFs have the best interfacial charge transfer and kinetic process among the materials evaluated for photocatalytic $CO_2$ reduction. The optimal catalytic activity was obtained in the catalytic reaction system, which provided a new direction for guiding and controlling the synthesis of new MOF materials.

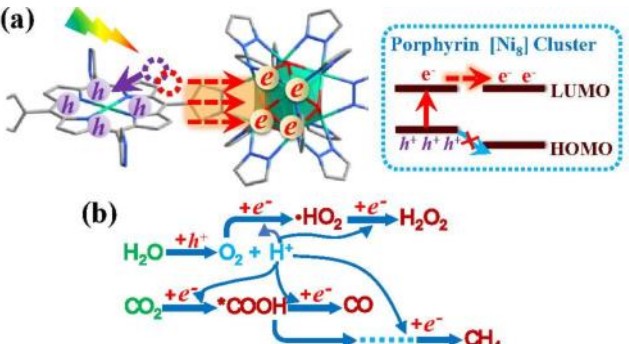

**Figure 6.** Proposed mechanism for $CO_2$ photoreduction by PCN-601 with **(a)** charge separation and transfer through a coordination sphere; **(b)** reaction pathways for the products. Reprinted with permission from Ref. [85]. 2020 Fang, Z.-B.; Liu, T.-T.; Liu, J.; Jin, S.; Wu, X.-P.; Gong, X.-Q.; Wang, K.; Yin, Q.; Liu, T.-F.; Cao, R.

### 2.5. Other Semiconductor Photocatalysts

Compared with commonly used photocatalysts (MHPs, MOFs, $TiO_2$, etc.), other efficient photocatalytic materials for the reduction of $CO_2$ need to be developed and studied. The conduction band position of SiC has a comparatively more negative potential, which can produce photogenerated electrons with stronger reduction ability for the photocatalytic reduction of $CO_2$. However, the synthesis of SiC in a high-temperature protective atmosphere is not conducive to the regulation of its nanostructure [86,87]. Layered double hydroxides (LDHs) such as Zn–Al LDH [88], Mg–Al LDH [89] and Zn–Cu–Ga LDH [90] have been used for the photocatalytic reduction of $CO_2$. Teramura et al. [91] synthesized a variety of LDHs with surface alkaline sites for the photocatalytic conversion of $CO_2$ to CO. Their activities tend to be higher than that of pure hydroxide. Graphite carbon nitride (g-$C_3N_4$) is a metal-free polymer material considered to be a promising visible-light catalyst [92,93]. Hsu et al. [94] used graphene oxide as a catalyst for the efficient photocatalytic conversion of $CO_2$ to methanol, and synthesized a graphene catalyst with an improved method to improve the activity. The yield of $CH_3OH$ was six times that obtained using pure $TiO_2$. The photocatalytic activity of highly porous $Ga_2O_3$ for the reduction of $CO_2$ was found to be more than four times that of commercial $Ga_2O_3$ without the need for additives [95]. The performance of porous $Ga_2O_3$ was improved by doubling the surface area and tripling the adsorption capacity. Tanaka et al. [96] used $H_2$ instead of $H_2O$ as a reducing agent to photocatalytically reduce $CO_2$ on $Ga_2O_3$, and the product was CO instead of $CH_4$. Notably, nearly 7.3% of the surface-adsorbed $CO_2$ was converted.

An effective strategy is to develop new semiconductor photocatalysts with visible-light response to improve the utilization rate of sunlight, so as to obtain higher photocatalytic activity. Zhou et al. [97] prepared $Bi_2WO_6$ square nanoplates by a hydrothermal method, and the product obtained by reducing $CO_2$ was $CH_4$. Figure 7 shows the result of $CH_4$ formation on $Bi_2WO_6$ photocatalysts. Cheng et al. [98] successfully prepared hollow $Bi_2WO_6$ microspheres, and methanol was obtained by reducing $CO_2$. Xi et al. [99] prepared $W_{18}O_{49}$ nanowires by a one-step liquid-phase method. Under visible-light irradiation, $CO_2$ was photocatalytically reduced in water vapor to generate $CH_4$. The average formation rate of $CH_4$ was significantly increased by using Pt and Au as cocatalysts.

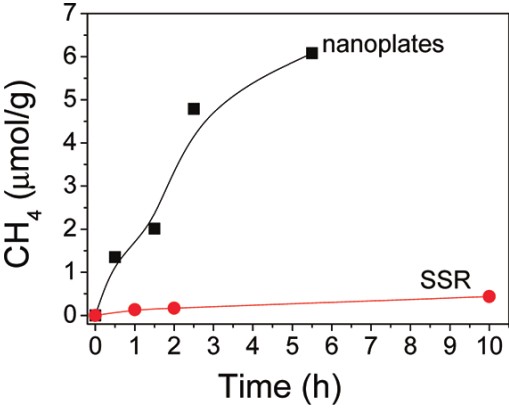

**Figure 7.** $CH_4$ yields on $Bi_2WO_6$ photocatalysts. Reprinted with permission from Ref. [97]. 2011 Zhou, Y.; Tian, Z.; Zhao, Z.; Liu, Q.; Kou, J.; Chen, X.; Gao, J.; Yan, S.; Zou, Z.

## 3. Principle of the Photocatalytic Reduction of $CO_2$

### 3.1. Process of the Photocatalytic Reduction of $CO_2$

Among many catalytic technologies, green and cheap semiconductor photocatalytic technology provides a new approach to the realization of this process. The inspiration for the semiconductor photocatalytic reduction of $CO_2$ comes from the photosynthesis of green plants in Nature. In Nature, plants absorb solar energy to convert $CO_2$ and $H_2O$ into organic matter and release oxygen to sustain their growth. Artificial photosynthesis using synthetic photocatalysts is used to simulate natural photosynthesis and reduce $CO_2$

into chemical products with high added value using sunlight. This is the most promising environment-friendly technology for $CO_2$ reduction at present.

Figure 8 shows a diagram of the mechanism of photocatalytic $CO_2$ reduction using semiconductor materials [100]. It can be seen from the figure that the photocatalytic reduction of $CO_2$ by semiconductor materials involves three basic processes. First, when the semiconductor photocatalyst is irradiated by incident light with an energy greater than or equal to its band gap (E.g.), the electrons in the valence band are excited and transition to the conduction band to form excited-state electrons with strong reduction ability. Holes with strong oxidation capacity equal to the number of electrons are left in the valence band, thus forming electron-hole pairs. The electrons and holes are then transferred to the surface of the semiconductor catalyst. Finally, the electrons reduce the $CO_2$ molecules on the surface of the catalyst, and the holes convert the $H_2O$ molecules on the surface of the catalyst into $O_2$. Most of the photogenerated electrons and holes recombine because the recombination rate of electrons and holes is much faster than the rate of surface redox reactions. Energy is released as light or heat, and only a small fraction of the electrons/holes can reach the surface of the catalyst to participate in the redox reaction. Therefore, it is very important to improve the separation efficiency of electrons and holes in the photocatalytic reduction of $CO_2$.

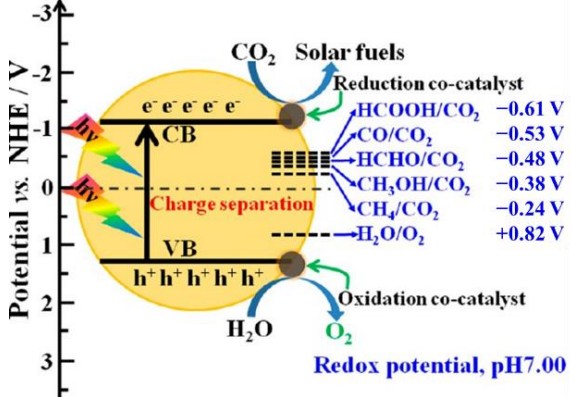

**Figure 8.** Schematic of the photocatalytic reduction of $CO_2$ by semiconductor materials. Reprinted with permission from Ref. [100]. 2016 Li, K.; Peng, B.; Peng, T.

### 3.2. Types of Photocatalytic $CO_2$ Reduction

The $H_2O$ molecule is an ideal reducing agent for the reduction of $CO_2$, providing protons to consume holes and producing oxygen and hydrogen during the reaction. At present, most photocatalytic $CO_2$ reduction reactions need to build reaction systems based on $H_2O$ molecules. The reaction system mainly includes two types, namely, gas-solid and liquid-solid systems. Figure 9 shows schematics of gas-solid and liquid-solid systems for photocatalytic $CO_2$ reduction.

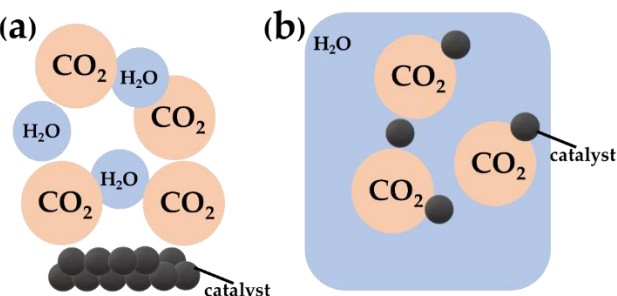

**Figure 9.** Schematics of gas-solid (**a**) and liquid-solid (**b**) systems for photocatalytic $CO_2$ reduction.

In a gas-solid system, the $H_2O$ molecule is present as water vapor and the catalyst is uniformly dispersed at the bottom of the reactor or uniformly coated on the substrate and placed at the bottom of the reactor. $CO_2$ gas flows into the reactor as a continuous stream or exists in the closed off-line system as a saturated gas. Zhang et al. [101] found that if $TiO_2$ nanotubes supported by Pt were used as photocatalyst, methane production increased when the concentration of water around the catalyst and the surface –OH group was relatively high. However, when Pt–$TiO_2$ nanoparticles were used as catalyst, the $CO_2/H_2O$ ratio had no effect on methane production. This indicates that the adsorption energy of water molecules on the catalyst surface affects the activity of the gas-phase photocatalytic system. Due to the microporous structure and surface properties of the materials, the adsorption of $CO_2$ and $H_2O$ on the active sites leads to different photocatalytic processes and the formation of different products [102].

In a liquid-solid system, the catalyst is suspended in an aqueous solution. Before the reaction, $CO_2$ gas is bubbled into the water to reach the saturation state [103]. The yield of product when using $H_2O$ as reducing agent is very low. It was found that adding triethanolamine or isopropanol into the solution consumes holes to improve the efficiency of photocatalytic $CO_2$ reduction. Kaneco et al. [104] used $TiO_2$ to photocatalytically reduce $CO_2$ in isopropanol solution. The $CO_2$ was reduced to methane, and isopropanol was oxidized to acetone. Another factor with a great influence on the reaction system is the suspension pH. Alkaline substances (NaOH, $NaHCO_3$, etc.) are usually added to the solution to improve the solubility of $CO_2$ and promote the reduction of $CO_2$ [105]. The pH of the suspension decreased from about 14 to 7.5, indicating that the solution mainly contained $HCO_3^-$ and a small amount of $CO_3^{2-}$, which could have accelerated the photocatalytic reduction of $CO_2$. The $OH^-$ ions in alkaline solution can be used as strong hole-trapping agents to improve photocatalytic activity. Wu et al. [106] significantly increased the yield of $CH_3OH$ by adding NaOH solution to the photocatalytic system solution. In addition, the addition of electrolytes increased total production as well as C2 products such as ethanol and acetaldehyde.

The difference between the gas–solid and liquid–solid systems mainly lies in the different concentrations of $CO_2$ and $H_2O$ on the surface of the catalyst; the products under different concentrations are also different. The disadvantage of gas-solid systems is that the contact area between reactants and catalyst is small, while the disadvantage of liquid-solid systems is that a large number of $H_2O$ molecules occupy adsorption sites for $CO_2$. Therefore, different optimization designs of catalysts are needed for different reaction systems in order to overcome the defects of the reaction system itself and maximize the efficiency of photocatalytic $CO_2$ reduction.

### 3.3. Reaction Conditions of Photocatalytic $CO_2$ Reduction

Photocatalytic reactions are usually influenced by external environmental and operational conditions, such as the wavelength range and intensity of light, the quality of the catalyst, the pH value of the solution, the pressure of $CO_2$, the reducing agent and the temperature. Hou et al. [107] studied the influence of different irradiation sources in the photocatalytic reduction of $CO_2$ with Au nanoparticle/$TiO_2$ catalyst, and generated different products such as $CH_4$, CO, $CH_3OH$ and HCHO. It was found that the photocatalytic reduction product of aqueous $CO_2$ by Au nanoparticle/$TiO_2$ catalyst under different excitation wavelengths ($\lambda$ = 532 nm, 254 nm and 365 nm) was $CH_4$. However, additional reaction products (including $C_2H_6$, $CH_3OH$ and HCHO) were observed under UV light (254 nm mercury lamp). The photocatalytic activity and selectivity can also be effectively improved by increasing the concentration of $CO_2$. Mizuno et al. [108] reported that with the increase of $CO_2$ pressure, the adsorption rate of hydrogen on the $TiO_2$ surface gradually exceeded that of $CO_2$. This adsorbed hydrogen began to form low-mass hydrocarbons such as $CH_4$ and $CH_2CH_2$. In gas-phase systems, increasing the pressure of $CO_2$ and water vapor can also promote the binding of reactants at the active sites of the catalyst, thus improving the photocatalytic activity of $CO_2$ reduction [101]. A further variable that must

be considered in the photocatalytic reduction processes is the system temperature. Fox and Dulay believed that a small temperature change had little effect on the photocatalytic reaction [109]; however, high temperature not only increases the collision frequency and diffusivity, but also increases the rate of thermal activation steps. Guan et al. [110] used a Pt-loaded $K_2Ti_6O_{13}$ photocatalyst combined with an Fe-based catalyst to reduce $CO_2$ and water vapor under sunlight and high temperature. It was found that the yield of HCOOH, $CH_3OH$ and $C_2H_5OH$ increased significantly with the increase of reaction temperature.

The crystal phase structure of the catalyst must also be considered. If the structure is the same but the exposed crystal plane is different, the photocatalytic $CO_2$ reduction performance will be different. Yamashita et al. [111] studied the photocatalytic $CO_2$ reduction by $TiO_2$ single crystal with rutile phase, exposed to (100) and (110) surfaces. They found that the activity of the (100) surface was significantly higher than that of the (110) surface, and $CH_4/CH_3OH$ products could be detected in the $TiO_2$ system exposed to the (100) surface. It can be concluded that optimizing the microstructure of catalytic materials can significantly improve the photocatalytic $CO_2$ reduction activity.

## 4. Conclusions and Perspectives

Since the first Industrial Revolution, with the increasing frequency of human activities and the massive burning of fossil fuels, the emission of $CO_2$ into the atmosphere has been increasing. This will lead to global warming, sea level rise, land desertification and other extreme climate and environmental problems. At the same time, the rapid development of industry has led to sharp increases in energy consumption worldwide. However, as non-renewable resources, fossil energy reserves are very limited. The development of new renewable energy sources is currently an urgent need of human society. One promising strategy for the mitigation of negative effects brought by $CO_2$ emissions is to simulate natural photosynthesis and use clean and sustainable solar energy to convert $CO_2$ into hydrocarbon fuels with added value; this could simultaneously alleviate environmental pollution and energy shortage in today's human society. However, because $CO_2$ is a symmetrical linear molecule and carbon is in its highest oxidation state, it has high thermodynamic stability, making the process of $CO_2$ reduction by visible light extremely challenging.

Many studies have been published on the development and design of efficient photocatalysts to achieve photocatalytic $CO_2$ reduction, but the photocatalytic reduction conversion efficiency is far from sufficient to meet the needs of practical application. This is usually caused by the high recombination rate of photogenerated electron-hole pairs and the poor adsorption activation capacity of $CO_2$ molecules in photocatalytic $CO_2$ reduction processes. In order to solve the bottleneck problem of low $CO_2$ adsorption activation capacity in the process of $CO_2$ photoreduction, it is necessary to modify the catalyst. There are many common catalytic material modification methods, such as morphology control, surface microstructure control and grain size control.

In the future, the research focus of semiconductor catalysts for the photocatalytic reduction of $CO_2$ should still be on improving the efficiency of photocatalytic reduction of $CO_2$. The main research direction should be the development of new semiconductor catalysts and optimized reaction conditions. In addition, the study of the reaction mechanism and type of photocatalytic reduction of $CO_2$ on semiconductor catalysts is helpful to find ways to improve the catalyst activity. Meanwhile, optimizing the reaction path is of great significance for the large-scale synthesis of catalysts in the future industrialization process. From an economic perspective, Co/Ni-based catalysts have the lowest cost among the catalysts ($TiO_2$ photocatalyst, Co/Ni-based catalysts, MHPs, MOFs and other semiconductor photocatalysts) mentioned in this review, while MHP catalysts are relatively expensive. Finally, in order to achieve green catalysis, the preparation cost of the catalyst must be considered, and the catalyst should be applied to the industrial photocatalytic reduction of $CO_2$ through improved reactors and other methods.

**Author Contributions:** Conceptualization, M.C. and Y.C.; formal analysis, P.G.; investigation, Y.C.; resources, M.C.; data curation, P.G.; writing—original draft preparation, Y.C.; writing—review and editing, L.X.; funding acquisition, M.C. All authors have read and agreed to the published version of the manuscript.

**Funding:** This research was funded by the National Natural Science Foundation of China (grant number 21976094, 22176100) and the National Key Research and Development Project 9 (grant number 2018YFC0213802).

**Conflicts of Interest:** The authors declare no conflict of interest.

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
