# Peer review of "Research Progress in Semiconductor Materials with Application in the Photocatalytic Reduction of CO2"

_catalysts, doi:10.3390/catal12040372_

Round 1
Reviewer 1 Report
Review Comments on: Research Progress in Semiconductor Materials with Photocatalytic Reduction of CO2 Applications
CO2 reduction is an important process that can aid in simultaneously reducing emissions and integrating renewable energy. Therefore, this review titled “Research Progress in Semiconductor Materials with Photocatalytic Reduction of CO2 Applications” which focuses on progress in the field can inspire new research directions. But there are certain issues that need to be addressed before its publication.
What is the state-of-the-art semiconductor material currently? I expected to see information about what is the best semiconductor materials currently in terms of conversion efficiency and selectivity of CO2 reduction?
The authors need to highlight the novel parts of their work, there are several reviews covering semiconductor materials for CO2 reduction. For example, Adv. Funct. Mater. 2019, 29, 1901825 (DOI: 10.1002/adfm.201901825) covers CO2 reduction mechanisms, different oxide catalysts for CO2 reduction together with various structural engineering strategies.
In the first sentence of the abstract, what do the authors mean by “…emission of non-renewable fossil fuels”?
Page 8, line 310, the text mentions “methanol” whereas the graph (Figure 7) talks about CH4.
There were several places where the authors just made a summary of what was done instead of commenting on the findings to guide readers. For example:
Page 7, Line 278-280 – The authors stated that “The optimal catalytic activity was obtained in the catalytic reaction system, which provided a new direction for guiding and controlling the synthesis of new MOFs materials.” Under what conditions was the test performed, what were the results in terms of yield or selectivity etc.
On page 9, line 371 – 377 The authors describe the influence of pH on the CO2 reaction system. The authors need to provide some insight about which pH or range of pH values produce the best results? Or how much of the alkaline substances (NaOH, NaHCO3) are added to obtain optimal results?
On page 10, line 394 to 396, the authors wrote: Hou et al. [107] studied the mechanism of using different irradiation sources in the photocatalytic reduction of CO2 with Au nanoparticle/TiO2-catalyzed and generated different products such as CH4, CO, CH3OH and HCHO.
While readers may find this information useful, there is no guidance on which different irradiation sources were used and how they affect the product? Which irradiation source led to CH4, CO, CH3OH, HCHO? The authors should provide a more detailed interpretation of the findings to guide the readers as to which was the most appropriate irradiation source in terms of yield, selectivity etc.
The statement on page 10, line 394 to 396 was immediately followed by a sentence starting with “However”. If “however” is used at the beginning of a sentence, it means the sentence contradicts the one immediately preceding it but I found that the two sentences are not even related. One talks about use of different irradiation sources whiles the other talks about “increasing the concentration of CO2”
Major work needs to be done to improve the English. The article was difficult to read, there were a lot of incomplete sentences, grammar mistakes and typos.
Author Response
see attached word file

Reviewer 2 Report
The review is quite interesting for readers of the Journal. However, the review says little about the drawbacks of one method or another. Nevertheless, it is known that any human activity (including CO2 utilization) leads to an increase in entropy (no one has canceled the second law of thermodynamics). Therefore, in addition to describing the merits of the methods, one should also scrupulously examine their shortcomings. Furthermore, the economic aspect is poorly reflected in the manuscript: it would be interesting to know how expensive TiO2 photocatalyst, Co/Ni-based catalysts, MHPs, MOFs and other materials are? It seems to me that it is clear to anyone that the implementation of photocatalytic reduction of CO2 will not be cheap. All these remarks, nonetheless, do not detract from the review itself, which is interesting and necessary. I recommend it for publication.
Author Response
see attached word file
